# Fatty acids and chlorogenic acid content in *Plectranthus edulis* root tubers

Tsehaynew Fetene [1,2], Minaleshewa Atlabachew [1]*, Hailu Sheferaw[1], Chaltu Reta[1], Kidanemariam Teklay Hilawea[1]

1 Department of Chemistry, College of Science, Bahir Dar University, Bahir Dar, Ethiopia, 2 Department of Environmental Health, Wollo University, Dessie, Ethiopia

* atminale2004@yahoo.com

**Data Availability Statement:** All relevant data are within the manuscript and its Supporting Information files.

## Abstract

This study quantified the fatty acid profile and total chlorogenic acid content of various Ethiopian cultivars of the Plectranthus edulis tuber, traditionally known as 'Agew Dinich'. Lipid extraction utilized the Folch method and the acid-catalyzed derivatization method to derivatize the fatty acids into fatty acid methyl ester (FAME) were used. Whereas maceration was used to extract chlorogenic acid from the fresh and freeze- dried tuber samples. Analysis revealed a total of thirteen fatty acids in all *P. edulis* samples, with nine classified as saturated and four as unsaturated. Palmitic acid was the most abundant fatty acid in *P. edulis* and accounted for 40.57%–50.21% of the total fatty acid content. The second and third most abundant fatty acids in the *P. edulis* sample were stearic and linoleic acids, which accounted for 8.38%–12.92% and 8.12%–11.28%, respectively. We reported chlorogenic acid for the first time in this potato species and found it to contain a concentration of 211± 4.2–300±24.7 mg/100g of dry weight basis when the determination was made using fresh samples. On the other hand, these samples yielded a chlorogenic acid concentration ranging from 115 ±8.6 mg/100g-175±3.9 mg/100g of freeze-dried powder samples. These findings suggest that *P. edulis* tubers could represent a significant dietary source of both chlorogenic acid and fatty acids.

## Introduction

Potato is the most consumed food crop in the world, after wheat and rice. Potato is also Ethiopia's fastest-growing primary food crop and a source of cash income for smallholder farmers. It is an essential crop for food security and a source of nutrients in underdeveloped nations and hence plays an important role in providing food security, revenue production, and employment opportunities [1]. Recently, the price of cereals has increased worldwide, and in Ethiopia, the price subsequently stabilized at a high level, whereas the price of roots and tubers remained relatively low during the entire food crisis. This requires the improvement of the seed potato systems operating in the country [2].

In terms of genetic resources, Ethiopia is one of the richest countries in the world. One of Ethiopia's most economically productive edible tuber crops is *P. edulis*, also locally known as

**Funding:** The funders had no role in study design, data collection and analysis, decision to publish, or preparation of the manuscript.

**Competing interests:** The authors have declared that no competing interests exist.

the Ethiopian potato. Its indigenous tuber crops are mostly grown in the highlands of the country. It's usually called' hunger crops' as it fills the shortage of food between the period before cereal crop harvest, from August to November. In the community, it is commonly advised as a special diet for those who are recuperating from illness and serves as a medicinal plant; eating the boiled root can avoid loss of appetite [3].

*P. edulis* is also one of the most important tuber-producing plants whose potential has not been fully investigated and realized. According to a researcher, *P. edulis is* known by several dissimilar vernacular names in different parts of Ethiopia: Dinich Oromo in the Oromia region, Wolayta Donuwa around the Wolayta zone, Gamo Dinich around the Gamo Gofa zone, Agew Dinch in the Awi zone, and Gurage Dinich around the Gurage zone of Ethiopia [4]. The plant has been grown in mid and high-altitude areas ranging from 1880 to 2200 meters above sea level. *P. edulis* is one of the most popular meals in the aforementioned districts or zones and is frequently offered to distinguished guests visiting a household. It is traditionally recommended as a special diet in the community for those recovering from illness, most likely due to its greater digestability, open-up appetite while eaten, and believed to have some medicinal properties [5]. It has a wide variety of adaptations and was previously commonly grown in the central, southern, western, northwestern, and southwestern parts of Ethiopia [6].

Oil seeds are excellent natural sources of lipophilic chemicals that can be used in cosmetic, medicinal, and biofuel products. It is worthwhile to look at the possible sources of lipids found in fruits and fruit byproducts [7].The lipid composition and chlorogenic acid composition of tuber plants are regarded as important parameters to distinguish between varieties and regions of origin. Despite the importance of compound class in *P. edulis*, no information is available on the total lipid, individual fatty acids, and total chlorogenic acid compositions of *P. edulis* of the Amhara region, particularly in the Awi zone. However, each potato species' quality and chemical composition differ depending on its cultivars, geographical origin, genotypic variation, agronomic factors, and environmental factors, including altitude, temperature, and soil [8].

Chlorogenic acids (CGA) are the main phenolic compounds. They are thought to have antioxidant capabilities, which may play a significant role in preserving food, tissues, and organs from oxidative degeneration. According to research, a diet rich in CGA compounds can help prevent a variety of diseases connected with oxidative stress, including cancer, cardiovascular disease, aging, and neurological disease [9].

Fatty acids originating from the diet have varying effects on the cardiovascular health of humans. Saturated fatty acids (SFA) have a somewhat favorable correlation with the risk of CVD, whereas trans fatty acids (TFA) have a strong and well-established link with the high risk of CVD. Poly- and mono-unsaturated fatty acids (PUFA and MUFA) generally lower the risk of cardiovascular disease (CVD) [10].The fatty acid composition of food is very important because lipids are one of the three major constituents of food. Their roles in biological tissues are: source of energy; components of biological membranes; a precursor for many different molecules [11]. Fruits, vegetable oils, seeds, animal fats, and fish oils are just a few examples of diverse sources of fatty acids [12]. Thus, monitoring the fatty acid composition of foods is an important process which can help maintain healthy life-style. However, most researchers in this area do not look at fatty acids and chlorogenic acids in *P. edulis* tubers collected from the Amhara region of the Awi zone, and its current production is declining to the extent of total extinction in several administrative regions.

Nowadays, *P. edulis* has been neglected in most parts of the regions where it is cultivated because consumers do not understand the important essential nutrients contained in the plant tuber, and hence it has not been promoted well. Therefore, the principal objective of this study was to identify and quantify the fatty acid and total chlorogenic acid constituents of Agew Dinch (*P. edulis*) from the Awi zone in the Amhara region. Thus, investigating its chemical

composition will help traders and nutritionists promote the plant in other parts of the region. If so, farmers who cultivate this tuber will benefit if the market chain is expanded. Furthermore, this study may add scientific information to the scientific community.

## Material and methods

### Equipment

The study utilized a variety of laboratory equipment and instrumentation, including a freeze dryer (SCANVAC, Germany), a platform shaker (ZHWY-334, Shanghai ZHICHENG), a centrifuge (model 800–1), plastic bags, a GC-MS (Agilent Technologies 7890B-5977A, China), an electrical girder (FW-100, high-speed universal disintegrator girders), an oven (universal hot air oven, New Delhi-110020, India), an electronic balance, a constant temperature and humidity incubator, a fume hood (Neuberger, type AZ 150), a digital inverter technology refrigerator (Samsung), a UV-Vis spectrophotometer (HACH DR6000, Linderberg drive Loveland USA), and Acro disc syringe filters.

### Reagents

The study employed a comprehensive selection of analytical-grade reagents and standards. A 99% pure pentadecanoic acid standard, a fatty acid compound, was utilized as an internal reference for the laboratory analyses. Additionally, the following high-purity chemical reagents were procured from reputable suppliers: Absolute, acetone-free methanol (99.00% purity) from Alpha Chemika. Chloroform (trichloromethane, $CHCl_3$, 99.8% purity) from Loba Chemie. Toluene (99% purity) from Blulux Laboratory. n-Hexane (99.9% purity, AR grade) from Pentokay Organy (India) Ltd. Sulfuric acid (98% purity) from a supplier in India. Anhydrous sodium sulfate from Blulux Laboratory. Sodium chloride (99.9% purity) from Blulux Laboratory. Chlorogenic acid standards (99.9% purity) from Sigma Aldrich.

### Sample collection and processing

The research was conducted under the approval and oversight of the research and community engagement vice dean office of the College of Science and Department of Chemistry at Bahir Dar University. During the 2022 crop harvest season, five samples were directly collected from farmers from five sub-districts of the Awi zone in the Amhara Region using a purposive sampling technique. Based on the information obtained from the Amhara Region Bureau of Agriculture, these five sub- districts are the zone's most significant *P*. edulis-producing sub-districts, which contribute voluminous *P*. edulis tubers to the market. From each sampling site, 2 kg of *P. edulis* samples were collected. The collected samples were packed and labeled in polyethylene plastic bags and were transported to the Bahir Dar University laboratory for further treatment.

The samples were washed with tap water, rinsed twice with distilled water, and further rinsed with deionized water to remove soil and dust particles and to minimize the load of microorganisms or other impurities. The cleaned tubers of *P. edulis* were sliced into small pieces with the help of a sharp stainless-steel knife for quick drying. The dried samples were ground using an electric grinder and sieved with a 0.5 mm pore-sized sieve (Fig 1). The flour was sealed in a double zip-lock plastic bag for crude fat and fatty acid analysis [13].

### Extraction of lipids

Lipids were extracted from the freeze-dried samples using the Folch method [4]. Briefly, 1.0 grams of *P. edulis* sample was taken in a test tube and mixed with 15 mL of chloroform and

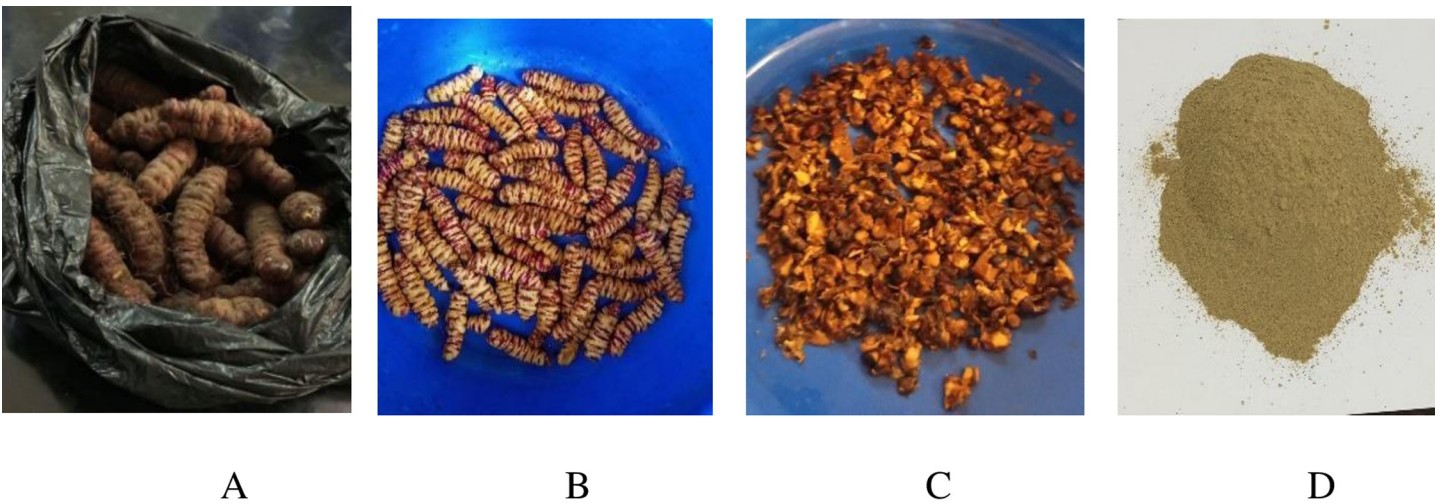

A                          B                          C                          D

**Fig 1.** Photo of *P. edulis* tuber (A), *P. edulis* wash(B) and *P. edulis* slices (C) *P. edulis* flour(D).

methanol (a 2:1 ratio v/v). The mixture was extracted for 36 hours on a platform shaker at 280 rpm then the extract was centrifuged, and the filtrate was taken (Fig 2). The lipid phase was separated with the aid of 2 mL of 0.73% aqueous sodium chloride, then the upper phase was removed by using a micropipette, and the lower phase (chloroform) layer containing the lipid was taken. The solvent was removed by letting the extract sit in a fume hood for two days, and the residue was reconstituted in 5.0 mL of toluene. The crude fat of each sample was calculated using the formula.

$$\%\text{Crude fat} = \left(\frac{W2 - W1}{WS}\right) * 100$$ where, $W_2$ = mass of the beaker with crude fat, $W_1$ = mass of empty beaker and, $W_s$ = mass of sample used.                Eq (1)

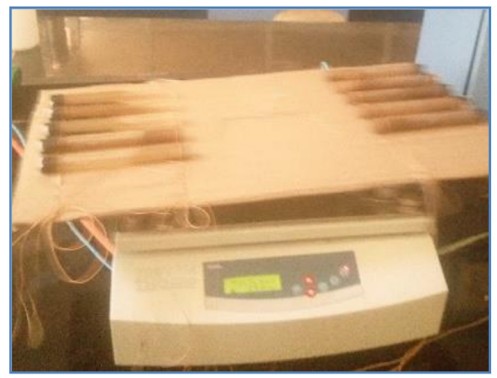 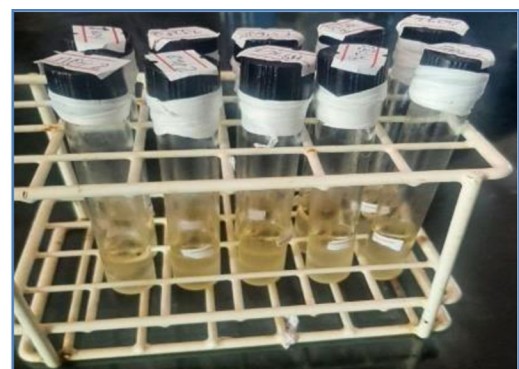

(A)                                              (B)

**Fig 2.** Extraction of lipids on platform shaker (A) and extracted lipids from P. edulis) (B).

## Derivatization of fatty acids

The acid derivatization method was used to convert the fatty acids to the corresponding methyl ester [14]. Briefly, a 2.0 mL portion of the lipid extract in toluene was spiked with 50 μL of 3.48 mg/ml Penta decanoic acid were allowed to react for 12 hours with 2.0 mL of 1% sulfuric acid in methanolic solution while being kept at 50˚C in an incubator. After that, the reaction mixture was treated with 5.0 mL of a 5% aqueous sodium chloride solution and extracted twice with 3 mL of hexane. After phase separation, the upper phase was taken away by using a micropipette (siphoning) and dried over anhydrous sodium sulfate, filtered with an Acro disc syringe filter, transferred into the vial, and analyzed and submitted to GC-MS analysis.

## GC-MS analysis

An Agilent Technologies 7890B gas chromatographic system equipped with an autosampler, a split spitless injector, and a mass spectrometer (Agilent Technologies 7890B-5977B) was used for GC-MS analysis of the fatty acid methyl esters. The following were the chromatographic conditions: 280˚C injector temperature, HP-5 MS UI- capillary column (30m x 250μm x 0.25m), temperature programming settings of 60˚C (held for 3 minutes), then ramped at 5˚C min-1 to 230˚C (held for 20 minutes), total run time 60.8 minute and helium as carrier gas at a flow rate of 1.0 ml min-1. The mass spectrometer (MS) was run with the following parameters: transfer temperature of 280˚C, scan range mz-1 of 40–600, ionization potential of 70 eV, and electron multiplier voltage of 935.9 volts.

## Identification of the fatty acids

The fatty acid methyl esters (FAMEs) were identified by comparing their retention times with respect to pure standard FAMEs purchased from Sigma Aldrich and analyzed under the same conditions [7].The identities of the detected fatty acids were determined by comparing the retention times and mass spectral fragmentation patterns of the fatty acids with the NIST-MS spectral library. In addition, a mixture of 19 reference standard fatty acid solutions was derivatized similar to the *P. edulis* extracts and analyzed by GC-MS. Then the retention time and fragment pattern of the sample's fatty acid and the reference standard were compared. The fatty acids used as a reference standard are Heptanoic acid, Nonanoic acid, undecanoic acid, Tridecanoic acid, pentadecanoic acid, 9-hexadecenoic acid, Hexadecanoic acid, Heptadecanoic acid, 9Z,12Z-Octadecadienoic acid, 9-Octadecenoic acid, Nonadecanoic acid, Octadecanoic acid, 5,8,11,14- Eicosatetraenoic acid, Adipic acid, Eicosanoid acid, 4, 7, 10, 13, 16, 19-Docosahexaenoic acid,13-Docosenoic acid, Adipic acid decyl 2,4-dimethyl-pent-3-methyl ester and Tricosanoic acid) compounds were analyzed. Out of nineteen mixed standard solutions seven were identified in *P. edulis* tuber samples. The identified mixed standards match with *P. edulis* samples are Nonanoic acid, 9- hexadecenoic acid, Hexadecanoic acid, 9Z,12Z, -Octadecadienoic acid, Heptadecanoic acid, 9- octadecenoic acid, and Eicosanoid acid.

## Quantification of fatty acids

Thirteen fatty acids, with relative percentages of peak area higher than 0.1%, were quantified accurately and used for the geographical origin comparison of the P. edulis sample types. These fatty acids: pelargonic (C9:0), capric acid (C10:0), lauric (C12:0), azelaic (C10:0), myristic (C14:0), palmitoleic (C16:1), palmitic (C16:0), linoleic (C18:2), margaric (C17:0), vaccenic (C18:1), arachidic (C20:0), oleic acid (C18:1), and stearic acid (C18:0) were determined relative

to the internal standard by using the following equation:[15].

$$\textbf{Concentration (in \%)} = \frac{indvidual\ peak\ area}{total\ peak\ area} \times \qquad\qquad \text{Eq (2)}$$

## Statistical analysis

The recorded data were expressed as Mean ± SD (standard deviation) and were statistically analyzed using the statistical software package SPSS (IBM Corp., USA). A one-way ANOVA was used to test the presence of significant differences in the mean content of fatty acids. In contrast, an independent sample test was performed to evaluate the presence or absence of significant differences in chlorogenic acid between fresh and freeze-dried samples.

## Determination of moisture content

To report the chlorogenic acid content of the samples on a dry weight basis, the moisture content was determined following the standard procedure. Briefly, 30 g of each *P. edulis* sample was accurately weighed in a crucible. The partially covered crucible was placed in the oven at 70°C for 6–12 hrs until the constant mass was obtained. Then the crucible was placed in a desiccator for 30 minutes to cool. The sample was reweighed after cooling and the percent of moisture was calculated.

## Extraction of chlorogenic acid from fresh and freeze-dried samples for UV-Vis analysis

The extraction was performed from both fresh and freeze-dried *P. edulis* samples with acidified water. The fresh *P.edulis* tuber was sliced into thin pieces [16]. Exactly 1.5 g of sample was placed in a 100-ml beaker and extracted with 50 ml of 1% acidified water by using a magnetic stirrer extractor at room temperature. After 90 minutes, the extract was decanted and filtered. Similarly, for freeze-dried samples, 0.4g of sample was extracted with the same solvent, time, instrument, and procedure as the fresh sample extract. The supernatant was taken from both fresh and freeze-dried sample extracts with a micropipette and analyzed by UV-Vis with a scan range of 200–800 nm [17]. A six-point calibration curve of chlorogenic acid (1 ppm, 2 ppm, 5 ppm, 10 ppm, 15 ppm, 20 ppm, and 25 ppm) was constructed to quantify chlorogenic acid content of the samples. All the standards were scanned from 200–800 nm and the maximum absorbance was taken to construct the calibration curve.

# Results and discussion

The moisture content of *P. edulis* sample ranged from 75.71% to 80.43%. This slight variation in the moisture content might be due to variation in the maturity stage of the tubers collected from the different districts of Awi Zone of Ethiopia.

## Chlorogenic acids

The UV-Vis spectra of chlorogenic acid are presented in (Fig 3). Looking at the figure, as concentration increases, the absorbance of the chlorogenic acid increases without shifting the position of the wavelength (325nm) of absorption.

The maximum absorbance value was taken for each concentration, and a calibration curve was constructed (Fig 3). Eq 1 shows the calibration curve equation generated from the calibration curve plot (Fig 3). The regression coefficient ($R^2$ = 0.9947) value generated from the

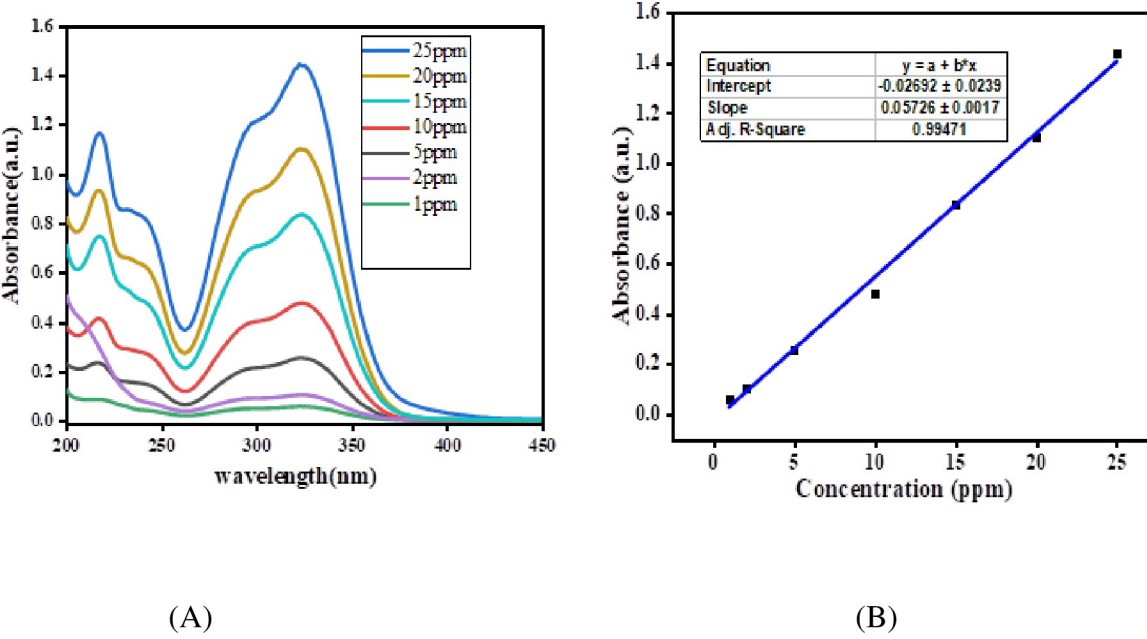

(A) (B)

**Fig 3.** UV-Vis spectra (A) and calibration curve (B) for the standard solution of chlorogenic acid.

calibration curve confirmed that good linearity was obtained over a concentration range of 1–25 ppm when plotted as a function of absorbance value at their λ max.

$$y = 0.05726\,x - 0.02692\,(\text{Chlorogenic acids}) \qquad\qquad \text{Eq (3)}$$

The UV-Vis spectra of the fresh and freeze-dried samples of each sample are presented in Figs 4 and 5.

The concentration of chlorogenic acid in the fresh sample was determined and presented in Table 1, and the minimum (211± 4.2 mg/100g) and the maximum (300±24.7 mg/100g) in a dry weight basis were observed in samples from Lemma and Gambisi, respectively. The reason for the variation could be attributed to several factors such differences in moisture content, the harvesting method, agricultural practices, or environmental factors (soil, altitude, and sun exposure).

When freeze-dried samples were used, the concentration of chlorogenic acid ranged from 115±8.6 to 175±3.9 mg/100g in dry weight basisof the dried sample (Table 2). It was noted that chlorogenic acid significantly decreased upon freeze-drying as compared to the fresh samples. Independent sample t-test analysis revealed a significant difference in chlorogenic acid content between the fresh and freeze-dried samples. The decrease in chlorogenic acid during the drying process as compared to the fresh samples might be the fact that the freeze-drying process removes the protective water through sublimation while also exposing the sample to atmospheric oxygen. Without water present, some phytochemicals, including some isomers of chlorogenic acid, may be more susceptible to degradation and oxidation pathways. Additionally, cell disruption during freezing and dehydration can expose the phytochemicals to endogenous plant enzymes that remain active even after freeze-drying, potentially leading to enzymatic breakdown of these compounds [18–21].

Reported studies indicated that the concentration of chlorogenic acid slightly decreased or decreased by half, and in some cases, the concentration of chlorogenic acid may also slightly increase or highly increase after freeze-drying the sample. For instance, freeze-drying slightly

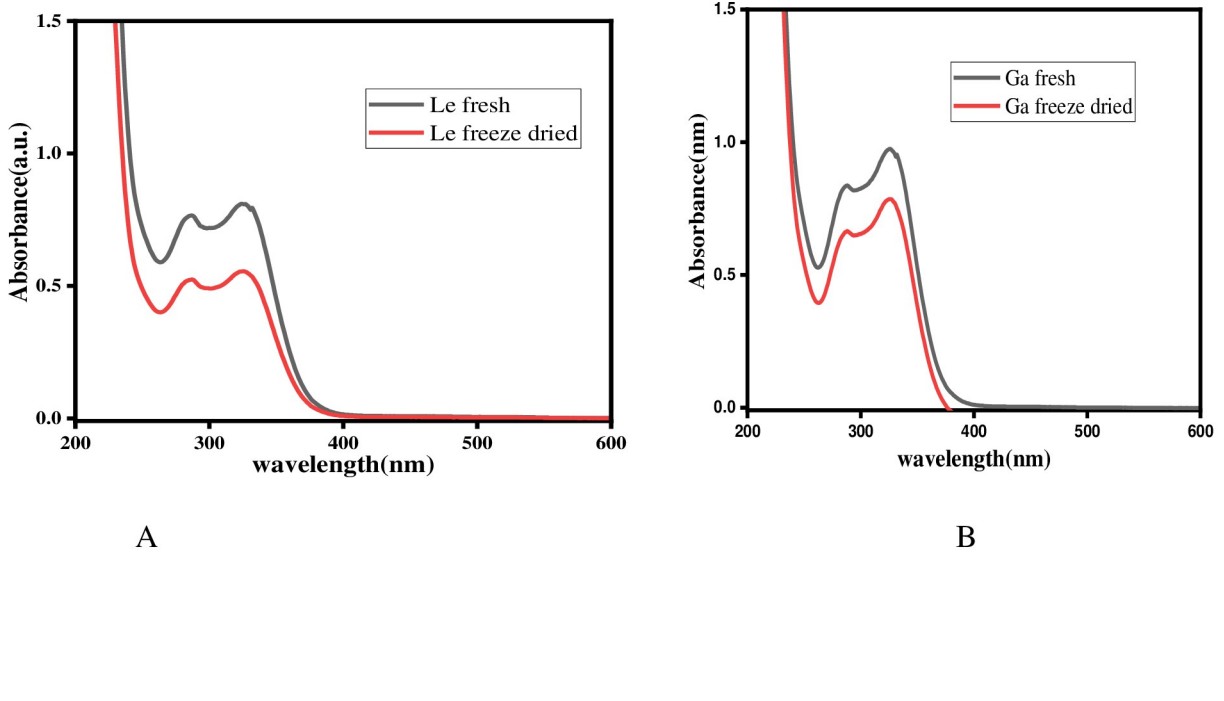

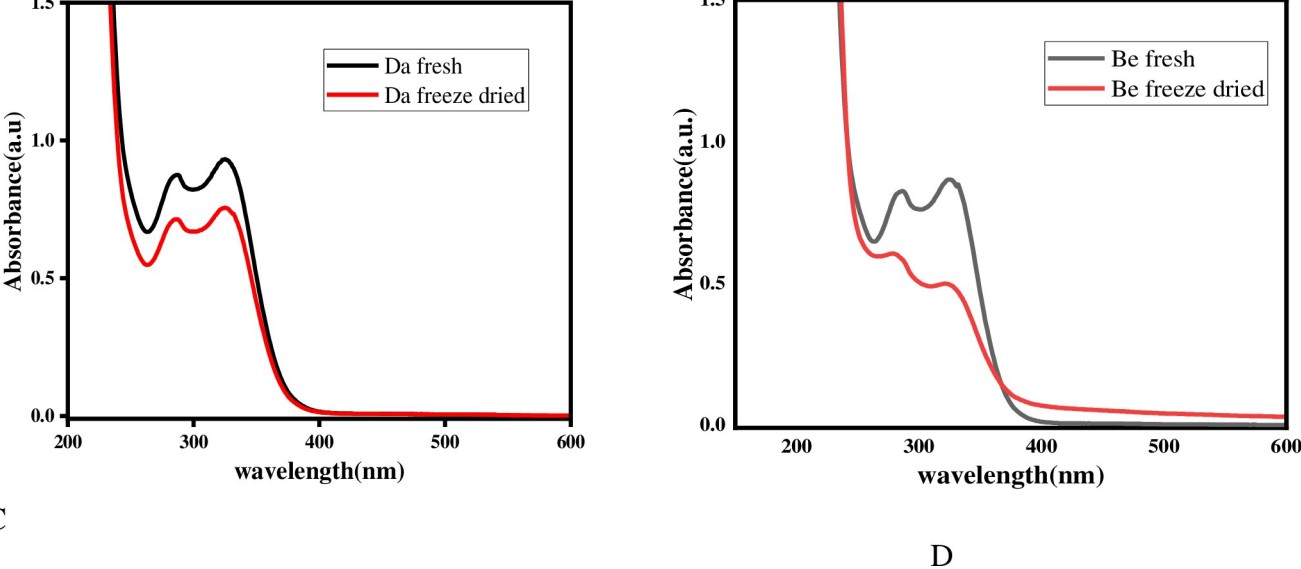

**Fig 4.** UV-Vis spectra of fresh and freeze-dried sample extracts from Lemma (A), Gambisi (B), Dangiya(C), and Betsena (D) sub-districts.

decreased the chlorogenic acid content in sweet potatoes from 154mg/g to 147mg/g [21]. For eggplant, freeze-drying was found to decrease the chlorogenic acid content by half—purple varieties went from 45% to 22%, and white varieties went from 97% to 49% [22]. Meanwhile, freeze-drying mulberry fruits slightly increased the chlorogenic acid content from 30mg/g to 33mg/g [23]. Grapes saw a highly increased chlorogenic acid content after freeze-drying, rising from 32.5mg/kg to 119.68mg/g [24]. Thus, the variation of the chlorogenic acid content with respect to freeze drying could depend on the nature of the plant material.

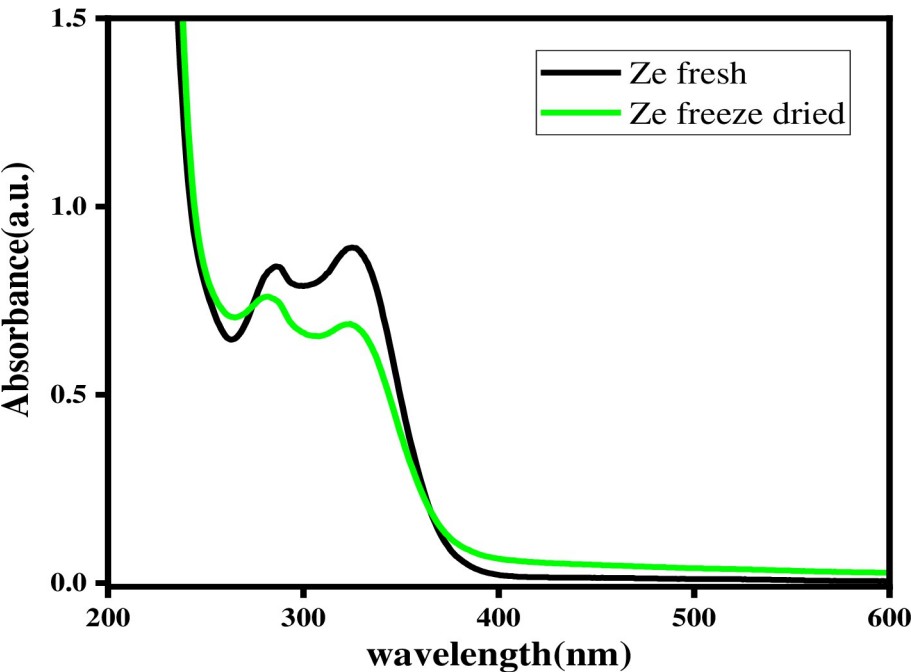

**Fig 5. UV-Vis spectra of fresh and freeze-dried sample extracts from Zengena sub-district.**

## Crude fat content

The crude fat content varied from 5.04% for Zengena to 6.47% for samples from the Lemma sub- district (S1 Table). According to [4] the crude fat content of *P. edulis* (Oromo Dinich) was determined with the range of (5.01% -10.87%). The current result was obtained in conjunction with the prior report since it falls within the range of the previous results (5.01%–10.87%) for *P. edulis* tuber.

Besides the internal standard to confirm the identity of the fatty acids in *P.edulis* samples 19 mixed standards (Heptanoic acid, Nonanoic acid, Undecanoic-acid, Tridecanoic acid, pentadecanoic acid, 9-Hexadecenoic acid, Hexadecanoic acid, Heptadecanoic acid, 9Z,12Z-Octadecadienoic acid, 9- Octadecenoic acid, Nonadecanoic acid, Octadecanoic acid, 5,8,11,14-

**Table 1. Concentration of chlorogenic acid (mg/100g) determined from fresh *P. edulis* samples.**

| No | Sample site | Moisture content | Absorbance | Mass of fresh Sample | Concentration of CGA on a fresh weight Basis | Concentration of CGA converted to dry weight Basis |
|---|---|---|---|---|---|---|
| 1 | Lemma | 77% | 0.81 | 1.5g | 48.7mg /100g | 211 ±4.2 |
| 2 | Gambisi | 80.4% | 0.97 | 1.5g | 58$mg$ /100g | 300±24.7 |
| 3 | Dangiya | 77.6% | 0.93 | 1.5g | 55.6$mg$ /100g | 248±27.6 |
| 4 | Betsena | 78% | 0.86 | 1.5g | 51.2$mg$ /100g | 235±4.9 |
| 5 | Zengena | 75.7% | 0.89 | 1.5g | 53$mg$ /100g | 219±21.9 |

NB: Concentration of chlorogenic acid on a dry weight basis was obtained by considering the moisture content.

**Table 2. Concentration of chlorogenic acid (mg/100g) determined from freeze-dried *P. edulis* samples.**

| No | Sample site | Absorbance | Mass of Dray sample | Concentration of CGA on a dry weight basis |
|----|-------------|------------|---------------------|--------------------------------------------|
| 1 | Lema | 0.56 | 0.4g | 128±10.8 |
| 2 | Gambisi | 0.78 | 0.4g | 175±3.9 |
| 3 | Dangiya | 0.75 | 0.4g | 173±7.1 |
| 4 | Betsena | 0.5 | 0.4g | 115±8.6 |
| 5 | Zengena | 0.68 | 0.4g | 155±8.2 |

Eicosatetraenoic acid, Adipic acid, Eicosanoid acid, 4, 7, 10, 13, 16, 19-Docosahexaenoic acid,13-Docosenoic acid, Adipic acid decyl 2,4-dimethylpent-3-methyl ester and Tricosanoic acid) compounds were analyzed. Out of nineteen mixed standard solutions seven were identified in *P. edulis* tuber samples. The identified mixed standards match with *P. edulis* samples are, Nonanoic acid, 9- hexadecenoic acid, Hexadecanoic acid, 9Z,12Z, -Octadecadienoic acid, Heptadecanoic acid, 9- octadecenoic acid, and Eicosanoid acid, to confirm the identity of the fatty acids (Table 3) and Fig 6.

## Identification of the fatty acids in the *P. edulis* samples

A total of thirteen fatty acids, nine saturated and four unsaturated fatty acids were detected in all of the P. edulis samples. A representative chromatogram of the sample is indicated in Fig 6 and the identified fatty acids are listed in Table 3. In Addition, the chromatogram of the reference standard solution is depicted in (S1 Fig).

**Table 3. The chemical name, retention time (RT), and means of identification of fatty acids determined in the P. edulis sample.**

| SN | CPD MAME | Chemical formula | Common name | Means of Identification | RT |
|----|----------|------------------|-------------|------------------------|-----|
| 1 | Nonanoic acid (S) | C9H18O2 | Pelargonic acid | NIST-MS+SD | 18 |
| 2 | Decanoic acid (S) | C10H20O2 | Capric acid | NIST-MS | 21 |
| 3 | Dodecanoic acid(S) | C12H24O2 | Lauric acid | NIST-MS | 26.1 |
| 4 | Nonane dioic acid (S) | C10H18O4 | Azelaic acid | NIST-MS | 26.7 |
| 5 | Tetra decanoic acid (S) | C14H28O2 | Myristic acid | NIST-MS | 32.3 |
| 6 | Pentadecanoic acid | C15H30O2 | (IS) | NIST-MS | 32.9 |
| 7 | 9-Hexadecenoic acid (Z)- | C16H30O2 | Palmitoleic acid | NIST-MS+SD | 34.5 |
| 8 | Hexadecanoic acid (S) | C16H32O2 | Palmitic acid | NIST-MS+SD | 34.9 |
| 9 | Heptadecanoic acid (S) | C17H34O2 | Margaric acid | NIST-MS+SD | 35.6 |
| 10 | 9,12-Octadecadienoic acid (Z, Z) (uns) | C18H32O2 | Linoleic acid | NIST-MS+SD | 38.1 |
| 11 | 9-Octadecenoic acid (uns) | C18H34O2 | Oleic acid | NIST-MS+SD | 38.2 |
| 12 | 11-Octadecenoic acid (uns) | C18H34O2 | Vaccenic acid, | NIST-MS | 38.3 |
| 13 | Methyl stearate (S) | C18H36O2 | Stearic acid | NIST-MS | 38.7 |
| 14 | Eicosanoic acid (S) | C20H40O2 | Arachidic acid | NIST-MS+SD | 42.3 |

NIST- MS = National Institute of Standards and Technology-Mass Spectroscopy; SD = standard.

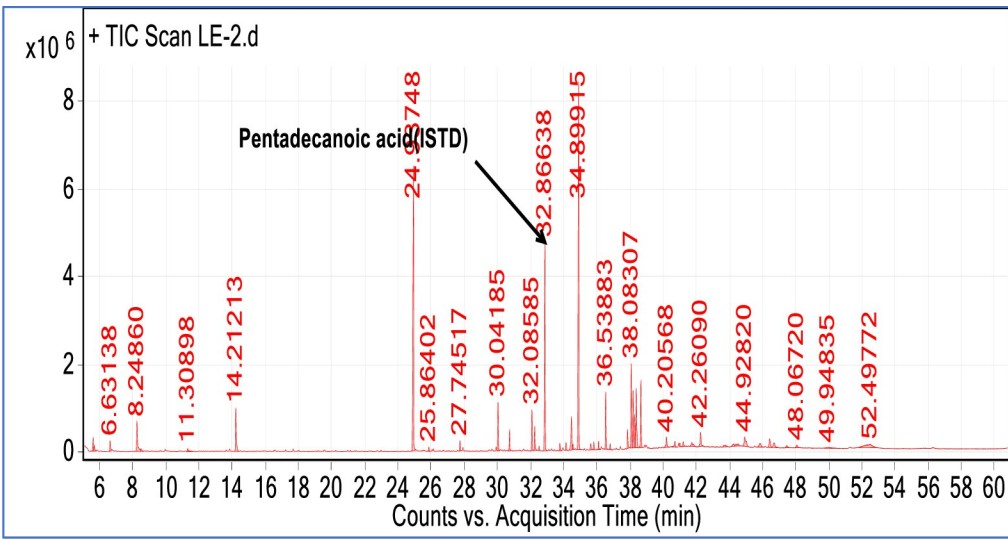

**Fig 6. A Typical GC-MS chromatogram of (*P. edulis*) extract, indicating the thirteen detected fatty acids.**

## Fatty acid profile of *P. edulis* at sub-district level

The fatty acid compositions of the *P. edulis* samples from the five sub-districts studied are given in Table 4. The total fatty acid contents of the *P. edulis* tuber ranged from 96.55–97.91% in samples in five sub-districts. The amount of palmitic, Stearic, and, Linoleic acids in *P. edulis* samples were 40.57±2.46–50.21±1.39%, 8.37±2.20–12.92±4.56%, and 8.12±0.29–11.28±1.11% of the total fatty acids, respectively. According to [25] the amount of linoleic and palmitic acids in two species of potatoes, (Solanum phureja and S. tuberosum) was indicated at 45–52% and 18–20% of the total fatty acids, respectively. According to [4] the predominant fatty acids in *P. edulis*(Oromo Dinich )present were linoleic followed by α-linolenic and palmitic acid. This demonstrates that there is no agreement on the current outcomes. In this study, palmitic acid was the most common fatty acid, followed by stearic and linoleic acids. Environmental variables caused agricultural and harvesting techniques in the Awi zone to differ from those in the rest of the country.,

The most common fatty acids present in *P. edulis* in the study of the five sub-districts are shown in Table 4. Among these, Pelargonic, Capric, Lauric, Azelaic, and margaric are present in trace amounts or modest concentrations of the total fatty acids and were identified in each sub-district and concentrations less than 1% of the total fatty acids. Among major fatty acids, Palmitic acid in *P. edulis* from all the five sub-districts: Lemma subdistrict has a higher concentration of palmitic acid than other Awi zone sub-districts. Stearic and linoleic acids were the second and third most prevalent fatty acids in the *P. edulis* sample, respectively. The observed variations in the fatty acid content between the *P. edulis* from the five different sub-districts of Awi zones can be ascribed to several factors, such as environmental growing conditions, agronomic factors, and harvesting conditions [26].

Regarding the agronomic factors, the *P. edulis* samples were processed by the same methods in all the sampling areas. However, because of the way the control and protection techniques for cultivation, the samples differed from farmer to farmer. Some farmers implemented best management methods such as soil preparation, adequate irrigation, crop fertilization, pest and disease management, and harvest and storage conditions s [27].

**Table 4. The mean, standard deviation, minimum, and maximum fatty acid percentages in *P. edulis* from five sub-districts.**

| | Five sub-districts of the Awi zone and the number of samples | | | | | | | | | | | | | | | |
|---|---|---|---|---|---|---|---|---|---|---|---|---|---|---|---|---|
| Fatty Acids | Betsena = 1 | | | Dangia, n = 1 | | | Zengena, n = 1 | | | Gambisi, n = 1 | | | Lemma, n = 1 | | | |
| | Mean±SD | Min | Max | Mean±SD | Min | Max | Mean±SD | Min | Max | Mean±SD | Min | Max | Mean±SD | Min | Max | P |
| Pelargonic Acid | 0.22±0.06 | 0.18 | 0.26 | 0.18±0.01 | 0.17 | 0.18 | 0.12±0.02 | 0.10 | 0.13 | 0.19±0.03 | 0.17 | 0.21 | 0.19±0.01 | 0.18 | 0.19 | A |
| Capric Acid | 0.12±0.01 | 0.10 | 0.13 | 0.19±0.02 | 0.18 | 0.20 | 0.12±0.00 | 0.12 | 0.12 | 0.18±0.06 | 0.14 | 0.22 | 0.22±0.06 | 0.18 | 0.26 | A |
| Lauric Acid | 0.41±0.04 | 0.39 | 0.44 | 0.42±0.05 | 0.38 | 0.45 | 0.27±0.01 | 0.26 | 0.28 | 0.37±0.03 | 0.35 | 0.39 | 048±0.04 | 0.46 | 0.51 | A |
| Azelaic Acid | 0.18±0.02 | 0.17 | 0.19 | 0.32±0.08 | 0.26 | 0.38 | 0.16±0.02 | 0.15 | 0.18 | 0.25±0.13 | 0.15 | 0.34 | 0.22±0.06 | 0.18 | 0.27 | A |
| Myristic Acid | 4.2±0.10 | 4.13 | 4.28 | 5.95±0.15 | 5.84 | 6.06 | 3.90±0.41 | 3.61 | 4.19 | 4.83±1.08 | 4.07 | 5.60 | 3.24±0.18 | 3.11 | 3.37 | A |
| palmitoleic acid | 9±2.00 | 7.59 | 10.42 | 5.92±1.84 | 4.61 | 7.22 | 7.07±0.18 | 6.94 | 7.19 | 6.25±1.91 | 4.90 | 7.60 | 6.21±0.46 | 5.88 | 6.53 | A |
| Palmitic Acid | 44±4.15 | 41.21 | 47.08 | 47.66±3.37 | 45.28 | 50.04 | 40.57±2.46 | 38.83 | 42.31 | 50.09±2.21 | 48.53 | 51.65 | 50.21±1.39 | 49.23 | 51.20 | A |
| margaric acid | 0.68±0.06 | 0.64 | 0.72 | 0.51±0.04 | 0.49 | 0.54 | 0.52±0.01 | 0.52 | 0.52 | 0.56±0.11 | 0.48 | 0.64 | 0.66±0.36 | 0.40 | 0.91 | A |
| Linoleic Acid | 9.59±1.74 | 8.36 | 10.81 | 8.12±0.29 | 7.92 | 8.33 | 11.28±1.11 | 10.49 | 12.06 | 8.45±0.83 | 7.87 | 9.04 | 10.16±1.23 | 9.29 | 11.03 | A |
| Oleic acid | 8.18±1.35 | 7.22 | 9.13 | 7.77±0.51 | 7.41 | 8.13 | 14.09±2.94 | 12.01 | 16.16 | 4.69±1.29 | 3.78 | 5.61 | 8.43±0.26 | 8.24 | 8.61 | A |
| Vaccenic Acid | 8.86±2.03 | 7.43 | 10.30 | 5.96±1.71 | 4.75 | 7.18 | 5.16±0.35 | 4.92 | 5.41 | 4.87±0.31 | 4.65 | 5.09 | 3.84±0.32 | 3.61 | 4.07 | A |
| Stearic Acid | 8.37±2.20 | 6.82 | 9.93 | 10.26±1.17 | 9.44 | 11.09 | 11.24±0.85 | 10.64 | 11.84 | 12.92±4.56 | 9.69 | 16.14 | 9.12±0.25 | 8.95 | 9.30 | A |
| Arachidic Acid | 3.22±0.24 | 3.05 | 3.40 | 3.01±0.45 | 2.69 | 3.33 | 2.32±0.09 | 2.26 | 2.38 | 3.62±0.48 | 3.29 | 3.96 | 3.29±0.13 | 3.20 | 3.38 | A |
| Total | 97.5±14.11 | 97.17 | 97.82 | 96.57±9.77 | 96.55 | 96.59 | 97.12±8.49 | 97.12 | 97.13 | 97.70±13.06 | 97.48 | 97.91 | 96.74±4.76 | 96.68 | 96.79 | A |

P = probability level, A = not significant at p > 0.05.

A statistical analysis using one-way ANOVA was performed to test the presence of significant differences in the fatty acid content of *P. edulis* between the five sub-districts of the Awi zone and it was noted that there was no significant difference in fatty acid content between and within a group of *P. edulis* samples sampled from five sub-districts (P>0.05).

The average fatty acid content (in %) present in *P. edulis* in the study of five different sub-districts of the Awi zone is shown in Fig 7. The Figure indicates that the contents of the average fatty acids in samples differ between subdistricts. Palmitic acid was discovered to be the main fatty acid in all sub-districts among the identified fatty acids. Stearic and linolic acids were the second and third most prevalent fatty acids in the *P. edulis* sample, respectively. The observed differences in the fatty acid content of *P. edulis* from the five different sub-districts can be attributed to a variety of factors, including land management systems, harvesting methods, agricultural practices, and environmental growing conditions [26].

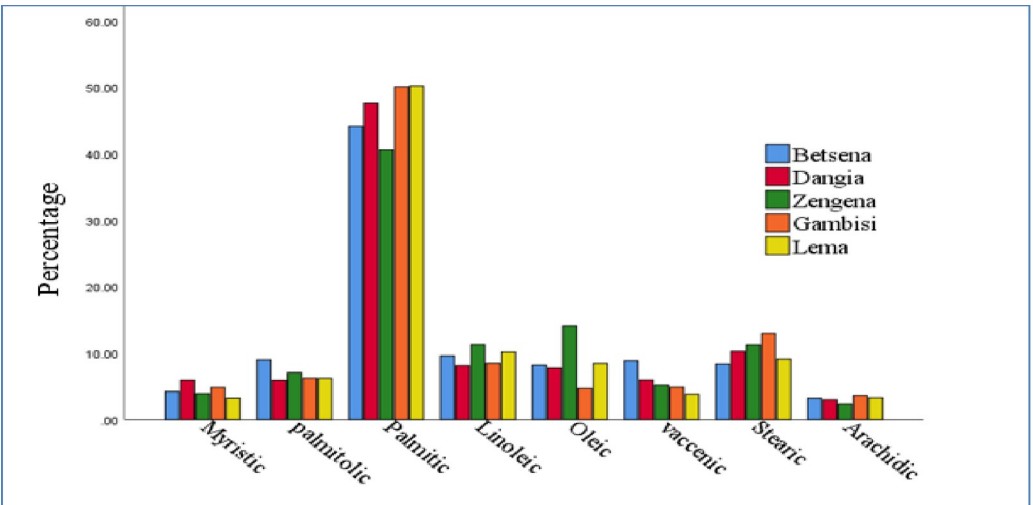

**Fig 7. Average fatty acid content of P. edulis sample in five districts of Awi zone.**

## Conclusion

The total crude fat and fatty acids in *P. edulis* samples representing selected sub-districts of the Awi zone were reported in this study. Among the determined fatty acids, Palmitic acid was found to be the major fatty acid in all samples. Stearic and linoleic acids were the second and third most prevalent fatty acids in the *P. edulis* sample, respectively. In addition, the studied tuber samples are rich in chlorogenic acid content. However, drying the samples could significantly reduce the chlorogenic acid content as compared to the fresh samples. In general, *P. edulis* tuber could be a good source of chlorogenic acid and fatty acids for individuals consuming the tuber. To have comprehensive phytochemical data on this *P. edulis* tuber, further studies are recommended, especially on the content of other phenolic acids, flavonoids, and amino acids.

## Supporting information

**S1 Fig. Chromatogram of the mixed standards of nineteen mixed standard.**
(TIF)

**S1 Table. Crude fat content of *P. edulis*.**
(DOCX)

## Acknowledgments

The authors would like to acknowledge Bahir Dar University of Ethiopia for providing laboratory facilities.

## Author Contributions

**Conceptualization:** Minaleshewa Atlabachew.

**Data curation:** Tsehaynew Fetene, Chaltu Reta, Kidanemariam Teklay Hilawea.

**Formal analysis:** Tsehaynew Fetene, Chaltu Reta.

**Investigation:** Tsehaynew Fetene, Chaltu Reta, Kidanemariam Teklay Hilawea.

**Methodology:** Minaleshewa Atlabachew, Kidanemariam Teklay Hilawea.

**Resources:** Chaltu Reta.

**Software:** Kidanemariam Teklay Hilawea.

**Supervision:** Minaleshewa Atlabachew, Hailu Sheferaw.

**Validation:** Minaleshewa Atlabachew, Chaltu Reta.

**Visualization:** Hailu Sheferaw.

**Writing – original draft:** Tsehaynew Fetene, Chaltu Reta.

**Writing – review & editing:** Minaleshewa Atlabachew, Hailu Sheferaw.

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
