## [Decision Letter · Decision Letter 0]

19 Apr 2024

PONE-D-24-04087Fatty acids and chlorogenic acid content in Plectranthus edulis root tubersPLOS ONE

Dear Dr. Atlabachew,

Thank you for submitting your manuscript to PLOS ONE. After careful consideration, we feel that it has merit but does not fully meet PLOS ONE’s publication criteria as it currently stands. Therefore, we invite you to submit a revised version of the manuscript that addresses the points raised during the review process.

We look forward to receiving your revised manuscript.

Kind regards,

James Nyirenda

Academic Editor

PLOS ONE

Journal Requirements:

"The authors would like to acknowledge Bahir Dar University of Ethiopia for providing laboratory

facilities. Mr. Tsehaynew Fetene is thankful to the Wollo university, Ethiopia, for sponsoring his

study"

5. Please ensure that you refer to Figure 1,2 and 4 in your text as, if accepted, production will need this reference to link the reader to the figure.

Reviewers' comments:

Reviewer's Responses to Questions

**Comments to the Author**

1. Is the manuscript technically sound, and do the data support the conclusions?

Reviewer #1: Yes

Reviewer #2: Partly

2. Has the statistical analysis been performed appropriately and rigorously? 

Reviewer #1: No

Reviewer #2: Yes

3. Have the authors made all data underlying the findings in their manuscript fully available?

Reviewer #1: No

Reviewer #2: Yes

4. Is the manuscript presented in an intelligible fashion and written in standard English?

Reviewer #1: Yes

Reviewer #2: No

5. Review Comments to the Author

Reviewer #1: The article is generally well written. The methods are described in detail and the results are presented logically in well-organized tables and graphs. The authors do perform basic statistical analysis to compare between the results.

Is the work technically sound? The research was done using standard methods that are replicable. The authors provide details on the sampling methods and how the samples were prepared. Detailed method of how the lipids and chlorogenic acid were extracted and analysed by GC-MS and UV vis spectroscopy (which authors can write in full) is provided.

Has the statistical analysis been performed appropriately and rigorously? The authors perform basic statistical analysis to compare the fatty acid content in the tubers from the different sub-districts. However the authors can also compare statistically the composition of chlorogenic acid between the samples from the different sub-districts and also between the fresh and freeze-dried samples.

Data availability: The authors provide the results logically and well organized tables and graphs. However the Supplementary data can include the detailed tables of the raw results. These results generally should be based on work done and not reporting other people's work.

Overall presentation: The manuscript generally flows well, however the authors need to check on a number of grammatical errors in the text, spacing between word and P. eludis should be written in itallics throughout the document. Authors should avoid starting sentences with "Because..."

Additional comments to the authors:

Page 1, in the abstract correct linolic acid to read linoleic acid.

Authors should avoid starting sentences abruptly. For example under introduction on page 2 paragraph 3, sentence “The authors also indicated that in different growth areas of Ethiopia, a dissimilar vernacular name is used for P. eludis”. It is not clear which authors are being referred to.

On page 3, the statements "the tuber can be consumed after boiling to maintain appetite. Furthermore, it is asserted that no matter how much is consumed, the stomach remains unaffected" are misplaced. Authors can consider rephrasing their statements to make them clear and also cite relevant literature to support the statements.

On page 7, the formula for crude fat analysis, brackets needs to be correctly placed. Some equations are numbered whereas some are not. This should be consistent.

On page 14, the authors can expand on the discussion of how freeze drying is affecting the chlorogenic acid content and if alternative methods of preservation can be recommended. Authors need to cite the necessary literature when they discuss how freeze drying is causing a change to the structure and composition of cells.

On page 21, the authors can highlight the key limitations of the study and make recommendations based on their findings or propose what future studies can be done to build on to their work.

Reviewer #2: !. The language should be improved including in the Abstract and Materials and methods sections.

2. Improve the presentation in the methodology section including indicating the molar concentrations of prepared solvents.

3. Improve on clarity of other equations

4. In results section, express the findings using mean +- stdev to appreciate variations

5. Maintain one format for Figures (Fig or Figure)

6. PLOS authors have the option to publish the peer review history of their article (what does this mean?). If published, this will include your full peer review and any attached files.

Reviewer #1: **Yes: **Evelyn Funjika

Reviewer #2: No

---

## [Author Response · Author response to Decision Letter 0]

3 Jun 2024

Replies to the technical comments of the Reviewers and the Editor

Manuscript ID: PONE-D-24-04087

We are very thankful to the reviewers and editor for their constructive criticism and valuable comments on our manuscript entitled "Fatty acids and chlorogenic acid content in Plectranthus edulis root tubers" and feel that they added a lot of value and clarity to the overall organization of the manuscript. We fully accepted all the issues raised and agreed that the issues raised should be addressed properly for better quality and acceptability of the manuscript. Therefore, we have revised our manuscript according to the editor and reviewers’ comments, and the changes that have been made in the revised manuscript, including texts and figures, are also highlighted with red font color. 

The replies to the Reviewers’ and editor’s comments are given carefully below:

Response to Editor’s comment

Comment #1: Please ensure that your manuscript meets PLOS ONE's style requirements, including those for file naming. The PLOS ONE style templates can be found at 

Authors’ reply: Ok, thank you for your constructive criticism and comments. The manuscript style already meets PLOS ONE’s style requirements, including file naming, affiliations, spacing, level of heading, figure, table, and reference citations.

Comment #2: In your Methods section, please provide additional information regarding the permits you obtained for the work. Please ensure you have included the full name of the authority that approved the field site access and, if no permits were required, a brief statement explaining why.

Authors’ reply: We have included a statement “The research was conducted under the approval and oversight of the research and community engagement vice dean office of the College of Science and Department of Chemistry at Bahir Dar University” under the section “ sample collection and processing”.

Comment #3: We note that the grant information you provided in the ‘Funding Information’ and ‘Financial Disclosure’ sections do not match. 

Authors’ reply: Ok, thank you for your constructive criticism, there is no funding information at all for this study.

Comment #4: Thank you for stating the following in the Acknowledgments Section of your manuscript: 

"The authors would like to acknowledge Bahir Dar University of Ethiopia for providing laboratory facilities. Mr. Tsehaynew Fetene is thankful to the Wollo university, Ethiopia, for sponsoring his study"

"The funders had no role in study design, data collection and analysis, decision to publish, or preparation of the manuscript." Please include your amended statements within your cover letter; we will change the online submission form on your behalf.

Authors’ reply: Sorry for these missed interpretations about funding information and financial disclosure, and the authors removed any funding-related text from the manuscript. We have included funding information in the revised cover letter for your assistance in updating it in the online submission form. 

Comment #5: Please ensure that you refer to Figure 1,2 and 4 in your text as, if accepted, production will need this reference to link the reader to the figure.

Authors’ reply: We have now cited Figures 1, 2, and 4 in the text as Fig 1, 2, and 4 respectively.

Response to Reviewer: 1

The article is generally well written. The methods are described in detail and the results are presented logically in well-organized tables and graphs. The authors do perform basic statistical analysis to compare between the results.

Authors’ reply: We thank your esteemed reviewer for the constructive feedback and appreciation of our study.

Is the work technically sound? The research was done using standard methods that are replicable. The authors provide details on the sampling methods and how the samples were prepared. Detailed method of how the lipids and chlorogenic acid were extracted and analyzed by GC-MS and UV vis spectroscopy (which authors can write in full) is provided.

Authors’ reply: We thank your esteemed reviewer for the constructive feedback and appreciation of our study.

Has the statistical analysis been performed appropriately and rigorously? The authors perform basic statistical analysis to compare the fatty acid content in the tubers from the different sub-districts. However, the authors can also compare statistically the composition of chlorogenic acid between the samples from the different sub-districts and also between the fresh and freeze-dried samples.

Authors’ reply: We thank the suggestion of our reviewer. We have now used an independent sample t-test to compare the variation of chlorogenic acid content in the freeze-dried and fresh samples. The detailed analysis data is included on page 14, in the last paragraph. 

Data availability: The authors provide the results logically and well-organized tables and graphs. However, the Supplementary data can include the detailed tables of the raw results. These results generally should be based on work done and not reporting other people's work.

Authors’ reply: We appreciate the feedback given, and we have removed a table containing reported data from the supplementary material document. The corresponding content from the removed table has been discussed in the manuscript (page 14, last paragraph).

Overall presentation: The manuscript generally flows well; however, the authors need to check on a number of grammatical errors in the text, spacing between word and P. eludis should be written in itallics throughout the document. Authors should avoid starting sentences with "Because..."

Authors’ reply: We have checked grammar and other issues, and corrected them in the revised 

Additional comments to the authors:

 Page 1, in the abstract correct linolic acid to read linoleic acid. 

Authors’ reply: We appreciate our reviewer's eagle eye view. We have corrected "linolic acid" to "linoleic acid."

 Authors should avoid starting sentences abruptly. For example, under introduction on page 2 paragraph 3, sentence “The authors also indicated that in different growth areas of Ethiopia, a dissimilar vernacular name is used for P. eludis”. It is not clear which authors are being referred to.

Authors’ reply: We have now significantly edited the paragraph to make it more clear. We have also cited proper references.

 On page 3, the statements "the tuber can be consumed after boiling to maintain appetite. Furthermore, it is asserted that no matter how much is consumed, the stomach remains unaffected" are misplaced. Authors can consider rephrasing their statements to make them clear and also cite relevant literature to support the statements. 

Authors’ reply: We have now rephrased the statement the tuber can be consumed after boiling to maintain appetite. Furthermore, it is asserted that no matter how much is consumed, the stomach remains unaffected" and cited a proper reference for it. The revised stament can be seen on page 3, first paragraph.

 On page 7, the formula for crude fat analysis, brackets need to be correctly placed. Some equations are numbered whereas some are not. This should be consistent. 

Authors' reply: We appreciate the comments. We have corrected all the issues our reviewer raised. The corrected section is as follows:

 formula for the crude fat is %Crude fat = ( (W2-W1)/WS)*100 where, W2 = mass of the beaker with crude fat, W1= mass of empty beaker and, WS= mass of sample used. ¬¬¬¬¬----eq (1).

 On page 14, the authors can expand on the discussion of how freeze drying is affecting the chlorogenic acid content and if alternative methods of preservation can be recommended. Authors need to cite the necessary literature when they discuss how freeze drying is causing a change to the structure and composition of cells. 

Authors' reply: We fully agree with this suggestion, The following information is included on page 14. “ The decrease in chlorogenic acid during the drying process as compared to the fresh samples might be the fact that the freeze drying process removes the protective water through sublimation while also exposing the sample to atmospheric oxygen. Without water present, some phytochemicals, including some isomers of chlorogenic acid, may be more susceptible to degradation and oxidation pathways. Additionally, cell disruption during freezing and dehydration can expose the phytochemicals to endogenous plant enzymes that remain active even after freeze-drying, potentially leading to enzymatic breakdown of these compounds. 

 On page 21, the authors can highlight the key limitations of the study and make recommendations based on their findings or propose what future studies can be done to build on to their work. 

Authors' reply: We have now suggested possible future studies to have compressive phytochemical content information in this tuber ( page 22, last paragraph).

Response to Reviewer: 2

 The language should be improved including in the Abstract and Materials and methods sections. 

Authors’ reply: We have thoroughly edited the language throughout the manuscript.

 improve the presentation in the methodology section including indicating the molar concentrations of prepared solvents. 

Authors’ reply: We have included the details on the concentration of solution prepared for this particular experiment.

 Improve on clarity of other equations 

Authors’ replyWe have edited the equation. We assigned equation number for each equation in the manuscript. 

 In results section, express the findings using mean mean ± stdev stdev to appreciate variations

Authors’ we have now reported our data in the result and discussion section as mean ± standard deviation

 Maintain one format for Figures (Fig or Figure)

Authors’ reply: The format for the Figures for this manuscript is maintained by Fig.

 PLOS authors have the option to publish the peer review history of their article If published, this will include your full peer review and any attached files.

Authors’ reply: “ yes”

---

## [Editor Report · Decision Letter 1]

7 Jun 2024

Fatty acids and chlorogenic acid content in Plectranthus edulis root tubers

PONE-D-24-04087R1

Dear Dr. Atlabachew,

We’re pleased to inform you that your manuscript has been judged scientifically suitable for publication and will be formally accepted for publication once it meets all outstanding technical requirements.

Kind regards,

James Nyirenda

Academic Editor

PLOS ONE
---

## [Editor Report · Acceptance letter]

27 Jun 2024

PONE-D-24-04087R1 

PLOS ONE

Dear Dr. Atlabachew, 

I'm pleased to inform you that your manuscript has been deemed suitable for publication in PLOS ONE. Congratulations! Your manuscript is now being handed over to our production team.

Kind regards, 

on behalf of

Dr. James Nyirenda 

Academic Editor

PLOS ONE